# Radioligands for Tropomyosin Receptor Kinase (Trk) Positron Emission Tomography Imaging

**DOI:** 10.3390/ph12010007

**Published:** 2019-01-03

**Authors:** Ralf Schirrmacher, Justin J. Bailey, Andrew V. Mossine, Peter J. H. Scott, Lena Kaiser, Peter Bartenstein, Simon Lindner, David R. Kaplan, Alexey Kostikov, Gert Fricker, Anne Mahringer, Pedro Rosa-Neto, Esther Schirrmacher, Carmen Wängler, Björn Wängler, Alexander Thiel, Jean-Paul Soucy, Vadim Bernard-Gauthier

**Affiliations:** 1Department of Oncology, Division of Oncological Imaging, University of Alberta, Edmonton, Alberta T6G 2R3, Canada; jjbailey@ualberta.ca (J.J.B.); eschirrm@ualberta.ca (E.S.); 2Division of Nuclear Medicine, Department of Radiology, The University of Michigan Medical School, Ann Arbor, MI, 48109, USA; amossine@med.umich.edu (A.V.M.); pjhscott@med.umich.edu (P.J.H.S.); 3The Interdepartmental Program in Medicinal Chemistry, University of Michigan, Ann Arbor, MI 48109, USA; 4Department of Nuclear Medicine, Ludwig-Maximilians-University of Munich, Marchioninistrasse 15, Munich 81377, Germany; Lena.Kaiser@med.uni-muenchen.de (L.K.); Peter.Bartenstein@med.uni-muenchen.de (P.B.); Simon.Lindner@med.uni-muenchen.de (S.L.); 5Program in Neurosciences and Mental Health, Hospital for Sick Children and Department of Molecular Genetics, University of Toronto, Toronto, ON, Canada M5G 0A4; dkaplan@sickkids.ca; 6McConnell Brain Imaging Centre, Montreal Neurological Institute, McGill University, 3801 University Street, Montreal, QC H3A 2B4, Canada; alexey.kostikov@mcgill.ca (A.K.); alexander.thiel@mcgill.ca (A.T.); jean-paul.soucy@mcgill.ca (J.-P.S.); 7Institute of Pharmacy and Molecular Biotechnology, University of Heidelberg, Heidelberg 69120, Germany; gert.fricker@uni-hd.de (G.F.); mahringer@uni-hd.de (A.M.); 8Translational Neuroimaging Laboratory, McGill Centre for Studies in Aging, Douglas Mental Health University Institute, Montreal, QC H4H 1R3, Canada; pedro.rosa.neto@gmail.com; 9Biomedical Chemistry, Department of Clinical Radiology and Nuclear Medicine, Medical Faculty Mannheim of Heidelberg University, 68167 Mannheim, Germany; Carmen.Waengler@medma.uni-heidelberg.de; 10Molecular Imaging and Radiochemistry, Department of Clinical Radiology and Nuclear Medicine, Medical Faculty Mannheim of Heidelberg University, Theodor-Kutzer-Ufer 1-3, Mannheim 68167, Germany; Bjoern.Waengler@medma.uni-heidelberg.de; 11Jewish General Hospital, Lady Davis Institute, Montreal, QC HT3 1E2, Canada; 12Azrieli Centre for Neuro-Radiochemistry, Research Imaging Centre, Centre for Addiction and Mental Health, Toronto, ON M5T 1L8, Canada; 13Department of Psychiatry, University of Toronto, Toronto, ON M5T 1R8, Canada

**Keywords:** tropomyosin receptor kinase, positron emission tomography, neurodegeneration, oncogenic fusions

## Abstract

The tropomyosin receptor kinases family (TrkA, TrkB, and TrkC) supports neuronal growth, survival, and differentiation during development, adult life, and aging. TrkA/B/C downregulation is a prominent hallmark of various neurological disorders including Alzheimer’s disease (AD). Abnormally expressed or overexpressed full-length or oncogenic fusion TrkA/B/C proteins were shown to drive tumorigenesis in a variety of neurogenic and non-neurogenic human cancers and are currently the focus of intensive clinical research. Neurologic and oncologic studies of the spatiotemporal alterations in TrkA/B/C expression and density and the determination of target engagement of emerging antineoplastic clinical inhibitors in normal and diseased tissue are crucially needed but have remained largely unexplored due to the lack of suitable non-invasive probes. Here, we review the recent development of carbon-11- and fluorine-18-labeled positron emission tomography (PET) radioligands based on specifically designed small molecule kinase catalytic domain-binding inhibitors of TrkA/B/C. Basic developments in medicinal chemistry, radiolabeling and translational PET imaging in multiple species including humans are highlighted.

## 1. Introduction

Tropomyosin receptor kinases (TrkA, TrkB, and TrkC) are transmembrane glycoproteins encoded by genes *NTRK1–3*, respectively. These kinases encompass extracellular domains (ECD) interacting specifically with endogenous neurotrophins as well as highly homologous intracellular tyrosine kinase domains (Figure 1A). Nerve growth factor (NGF) binds to TrkA, brain-derived neurotrophic factor (BDNF), neurothrophin-3 (NT-3), and neurotrophin-4 (NT-4) to TrkB, and NT-3 to TrkC [1,2,3]. Full length TrkB is a 140-kD transmembrane spanning protein, with an extracellular ligand-binding domain containing two cysteine clusters, leucine-rich repeats, and two immunogloblulin-like domains [4]. The intracellular domain encodes a tyrosine kinase domain that when activated transphosphorylates monomers of the TrkB dimer. When transphosphorylated, the Trks engage and phosphorylate their major substrates Shc, Phospholipase C γ1 (PLC-γ 1), and Fibroblast Growth Factor Receptor Substrate 2 (FRS2/SNT1) [5]. The TrkB locus also encodes four variants generated by alternative splicing, of which the most abundant is the 90-kD truncated TrkB.t1 isoform that lacks the kinase domain [6]. Trk signaling occurs primarily through Ras/Mitogen-Activated Protein Kinase 1 (MAPK1), Phosphoinositide-3-Kinase (PI3-K)/Akt and PLC-γ1 [4,7,8,9] and plays central roles in mediating neuronal survival and differentiation in the embryonic, postnatal, and mature peripheral (PNS) and central nervous system (CNS) [2,3]. Within the CNS, reduced expression, as well as abnormal and impaired signaling of Trk receptors, are associated with a plethora of neuropathologies, including ischemic brain injury, schizophrenia, Rett syndrome, depression, Parkinson’s disease (PD), and Alzheimer’s disease (AD) [10,11,12,13,14,15,16]. In AD, evidence from ex vivo experiments accumulated over the last two decades demonstrates reductions in full length catalytic TrkB/C receptor densities as well as a decline in TrkB/C neurotrophin signaling [17]. Direct evidence for the involvement of alterations in BDNF/TrkB signalling in AD is supported by studies indicating that TrkB levels are profoundly decreased in the hippocampus, frontal and temporal cortex of patients with Alzheimer’s [18], although apparently not in the parietal cortex [19]. Furthermore, progressive loss of TrkA, B, and C in basal forebrain cholinergic nuclei is well correlated with the clinical progression of AD [20]. Treatment with agonists of the BDNF/TrkB system of transgenic mouse models of AD increases dendritic spines in the hippocampus and cortex, inhibits neuronal apoptosis and neurodegeneration, and improves spatial memory performance [21,22,23,24,25]. These findings support the use of agents that activate BDNF/TrkB for treating AD [26]. Yet, the in vivo relevance or the spatiotemporal evolution of such changes, as well as the relationships which may exist with known neuropathological hallmarks of neurodegeneration such as plaque deposition and neurofibrillary tangle (NFTs) aggregation in AD—both of which can be visualized using diagnostic positron emission tomography (PET)—are unresolved questions. The potential importance of perturbations of Trk expression, activity and signalling in neurodegeneration hence raises the question of whether these molecular targets can in turn be imaged in vivo and non-invasively using PET. Another potential application for Trk-targeted radioligands stems from the renewed and rapidly growing interest in anti-Trk therapy for cancer. Indeed, important advances in recent years have been made in the treatment of patients with *NTRK* fusion-positive cancers in basket trials using pan-Trk inhibitors [27]. In parallel, remarkable progress has also been achieved in the development of selective pan-Trk and TrkA subtype-selective tyrosine kinase inhibitors (TKIs). *NTRK* fusions are found at low frequency in a number of common cancers and at a relatively high frequency in rare neoplasms—amounting to about 1500–5000 patients with *NTRK* fusions-positive diseases per year in the United States. Current clinical trials assessing *NTRK* fusion-positive patients inherently rely on tumour biopsy (which may not be always achievable) followed by next generation sequencing or fluorescence in situ hybridization for fusion detection. The use of Trk-targeted PET imaging in early clinical stages to assess receptor occupancy, dosing regimen, and *NTRK* fusion-positive status, or to monitor treatment response in place of sequential tumour biopsy may be both achievable and desirable—as previously done with other molecular targeted TKI therapies [28]. 

Until very recently, suitable imaging lead compounds or quantifiable non-invasive techniques to measure spatiotemporal fluctuations of TrkA/B/C levels have been unavailable. To address this, we undertook in 2014 the task of identifying structural determinants which would enable TrkA/B/C PET imaging. To this end, we developed structurally diverse Trk radiotracers and inhibitor libraries with various levels of potency and kinome selectivity, both from type I and type II inhibitor classes, and exploited diverse radiochemical approaches using carbon-11 and fluorine-18. While our primary objective has been non-oncological neuroimaging in the context of neurodegeneration and most results gathered thus far aimed at meeting this objective, we recognize that with the recent clinical oncological breakthrough in Trk inhibitor therapy comes a clear need for reliable and non-invasive assessment of Trk status in cancer therapy trials. In this short review, we describe the rational design and development of first-in-class Trk-targeted TKI PET radiotracers and delineate imaging validation obtained with these molecular probes to date.

## 2. The Development of Trk Radioligands for PET Imaging

### 2.1. Binding Site Considerations

PET radionuclides decay by emission of a positron, which in turn annihilates with a nearby electron, generating two gamma rays of 511 keV (conversion of the positron’s and electron’s mass into energy). These two gamma rays, emitted in the opposite directions, can then be detected by an outside PET camera, revealing the position of the annihilation events with sufficient spatial resolution (from submillimeter to millimeter in preclinical and clinical settings respectively). Under the assumption of a sufficient tissue target receptor concentration (Bmax), an ideal radiotracer, in this context, needs to meet certain criteria which include: (1) high radioligand concentration in the tissue of interest, (2) radiotracer equilibrium conditions are reached, (3) lack of interfering radiometabolites and (4) high on-target selectivity. Beyond requiring careful studies of possible radiometabolites, this also highlights the importance of targeting suitable domains of a molecular target. This is especially relevant in the case of Trk where various isoforms, splice variants or fusions proteins may be co-expressed within a single tissue in different clinical contexts. For example, in addition to the full length TrkA/B/C, a number of truncated isoforms lacking the intracellular kinase domain have been characterized for TrkB (Figure 1B). Within the brain, the protein expression levels of the truncated isoforms of TrkB present (TrkB.T1 and TrkB.T2) far exceed that of full length TrkB [29]. A radiotracer binding the TrkB extracellular domain (ECD) would consequently map the sum of all TrkB receptors in the brain including the truncated isoforms which as yet have no known reported association with neurodegeneration or as oncogenic drivers. A PET signal originating from such a radiotracer would fail to differentiate catalytically-incompetent truncated from full-length kinase domain-containing receptors and would be unable to engage targets which lack cognate ectodomains such as the many NTRK fusion proteins which are of prime clinical interest (Figure 1B). More pragmatically, another reason to avoid targeting ECDs of Trk for imaging stems from the lack of suitable compounds. Indeed, while the study of nonprotein Trk ECD-binding compounds largely predates the advent of Trk-targeted TKIs, very limited progress has been achieved due to the inherently challenging druggability of the ectodomains compared to the kinase domains of TrkA/B/C and the poor drug-like properties of such compounds. These limitations have been exemplified by our imaging study of radiolabeled 7,8-dihydroxyflavone (7,8-DHF) [30]. This work was based on the initial report by Jang et al. suggesting that 7,8-DHF binds with high affinity to the ECD of TrkB and exhibits agonistic activity in neurons [31]. From this initial work, other more active flavones were described including fluorinated analogs which caught our attention as potential starting points for ^18^F-PET tracer development [32,33]. Yet, using synthesized or commercially available described flavones we were unable to reproduce any basic biochemical effects in cells (unpublished data). In line with our findings, Boltaev et al. more recently unambiguously demonstrated the lack of direct interaction or TrkB-driven effect of such compounds [34]. In addition, we observed poor in vivo profiles in rats when investigating the radiolabeled isotopologs of these compounds for imaging (including 2-(4-[^18^F]fluorophenyl)-7,8-dihydroxy-4H-chromen-4-one). Preclinical imaging revealed low-radioactivity overall brain uptake with a maximum standard uptake value (SUV) of 0.64 (whole brain) followed by fast elimination from the brain. These radioligands also were rapidly eliminated from the plasma compartment by hepatic metabolism with an estimated plasma half-life under 4 min suggesting a fast and extensive phase II metabolism and suggesting overall poor druglike properties, as it is known for hydroxyflavones more generally [35,36]. Interestingly, in vitro autoradiography with rat brains revealed apparent specific and BDNF-competitive binding reminiscent of ^125^I-BDNF binding topology. Yet, we surmise that this binding/blocking effect may have been a consequence of covalent reactivity to nucleophilic cysteines or other non-selective thiol-based reactivity as previously described for pan assay interference flavonoids [37]. Compared to putative ECD binding compounds, inhibitors targeting the ATP binding site of the cytoplasmic Trk kinase domains have been well characterized both functionally and structurally and have been reported alongside detailed crystallographic data enabling derivatization [38]. Targeting the kinase domains however comes with the challenge of selectivity, firstly within the kinome and secondly from within the Trk kinase family itself (TrkA, TrkB and TrkC)—TrkA/B/C display 72–78% and 95–100% sequence identity in their kinase domain and ATP binding sites, respectively. We have strived to detail kinome selectivity using comprehensive kinase screening whenever possible. With the exception of a few compounds, all radiolabeled inhibitors display pan-Trk activities. Yet, owing to the high expression levels of TrkB/C in the CNS compared to TrkA [39,40,41,42,43], and given that the PET signal is driven by target density and ligand affinity, it is to be expected that CNS PET using pan-Trk radioligands will nevertheless detect nearly exclusively TrkB/C-based signal. Another important note pertains to the fact that Trk kinases can adopt both DFG-in (active, targeted by type-I inhibitors) and DFG-out (inactive, targeted by type-II inhibitors) conformations (Figure 2). Conformational changes upon DFG triad rearrangement allow for the access of a deep pocket which can be occupied favorably by lipophilic moieties (Figure 2B,C). Type II Trk inhibitors targeting DFG-out conformation thus also present longer residence times compared to type-I inhibitors. To study potential benefits from each binding modes, we have developed both type-I and II radioligands. The following paragraphs give an account on our Trk radiotracer development program in chronological order progressing towards first in human PET imaging from preclinical research. 

#### 2.1.1. The 4-Aza-2-oxindole Radioligands

PET tracers for neuroimaging have significantly higher chances of achieving high specific binding in brain when displaying certain physicochemical properties. Parameters such as rapid Blood Brain Barrier (BBB) permeation and neuron cell penetration as well as low non-specific binding are mostly determined by the compound’s lipophilicity (represented as clogD _7.4_ value, assuming passive diffusion through the cell membrane), reduced susceptibility to active efflux, molecular weight (<350 Da), favorable neuroreceptor binding potential (BP = B_max_/K_d_ >10), topological surface area (TSPA) of 30–80 Å^2^, and hydrogen bond donor ability (HBD) of >1. These general factors as well as specific properties tend to characterize successful neuroimaging PET tracers [44,45,46,47]. With these parameters in mind, we first opted for an inhibitor belonging to the family of 3-arylideneindolin-2-one Trk inhibitors. The 4-aza-2-oxindole inhibitor GW441756 [48] was screened as a compound favorably fulfilling the above mentioned requirements of a potentially successful candidate for brain imaging and was consequently labeled with carbon-11 to provide the radioactive isotopolog for autoradiography and PET imaging (half maximal inhibitory concentration (IC_50_); TrkA = 29.6 nM, TrkB = 6.7 nM, and TrkC = 4.6 nM, Figure 3) [49]. Even though the radiolabeling was straightforward and high yielding, the targeted *Z* isomer, which based on molecular modeling was found to be the active isomer, could not be obtained in pure form after purification of the crude radiolabeling solution. An *E/Z* 1:1 mixture was isolated under all conducted labeling conditions as a result of rapid in situ isomerisation either during the radiosynthesis or during the HPLC purification procedure, reducing the amount of high affinity Trk tracer *Z*-[^11^C]GW441756 by 50%. The inevitable contamination of the *E*-isomer with Z-[^11^C]GW441756 significantly compromised the merit of this tracer as a potential Trk PET imaging agent, not only from an imaging quality perspective (50% of the PET signal will be the result of the non-specific binding of the Z-isomer) but also from a clinical translational point of view. Autoradiography of the mixed isomers on coronal rat brain sections revealed a displaceable binding in accordance with the ubiquitous distribution of all TrkB/C subtypes. The in vivo evaluation of [^11^C]GW441756 was performed in Sprague–Dawley rats to evaluate BBB penetration and in vivo biodistribution. In baseline scans the tracer rapidly entered the brain peaking at a SUV of 2 after 30 s, after which the compound was rapidly eliminated from the brain over 30 min. Tracer distribution was uniform in accordance with ubiquitous TrkB/C expression in the rodent brain. A blocking study using unlabeled GW441756 pre-treatment was inconclusive as to whether the tracer specifically engages the CNS TrkB/C in vivo, however specific binding was indicated from blocking significant lung uptake of the tracer which may be linked to TrkB specific binding in pulmonary tissue [50]. An ^18^F derivative bearing an ^18^F-fluoroethyl group at the 5-hydroxy moiety of a new GW441756 derivative, although being highly selective over other kinases and displaying generally favorable in vitro properties (low IC_50_ values in the nanomolar range for all Trk subtypes), showed unexpected susceptibility towards CYP450 metabolism in vitro in contrast to the original GW441756. These findings were corroborated by rapid formation of brain penetrating metabolites and no observable blocking upon pre-treatment with non-radioactive GW441756 in an animal PET experiment in rats. It can be concluded from our data that radioligands for pan-Trk imaging with PET, derived from the 3-indolydene 4-aza-2-oxindole scaffold despite being selective and highly potent inhibitors of Trk, do most probably not lend themselves towards PET tracer development and clinical translation. The main reason is the inevitable presence of the low affinity *E*-isomer of [^11^C]GW441756 due to the rapid reaching of a photostationary state, a physicochemical reality that most definitely is not limited to GW441756 but rather extends to all structurally similar derivatives. However, both tracers displayed high specific binding in TrkB expressing neuroblastoma tumour sections in vitro providing a proof of principle of the utility of such radioligands for oncological imaging.

#### 2.1.2. 2,4-Diaminopyridmidine and Quinazoline-Based Radioligands

In order to explore the effect of targeting the DFG-out Trk kinase conformation, especially with regard to kinetics [51], we also developed type II radiolabeled inhibitors. In general, type II Trk inhibitors tend to possess mostly elongated structural features, higher molecular weight and HBD count than type-I inhibitors, which may be detrimental for BBB permeation but inconsequential in the context of peripheral imaging. So far we have developed two distinct type-II radiolabeled inhibitors belonging to the class of 2,4-diaminopyridines [52] and quinazoline [53] (Figure 3). Analyses of large kinase inhibitor sets identified GW2580, an orally active dual Trk/colony-stimulating factor-1 receptor (CSF-1R) inhibitor (*K*_d_; CSF-1R = 2 nM, TrkA = 630 nM, TrkB = 36 nM, and TrkC = 120 nM), as one of the most kinome selective inhibitors known [54]. Based on this attribute, and the overall promising preclinical ADME (absorption, distribution, metabolism, and excretion) of this inhibitor, we undertook to identify a radioligand based on this scaffold. We rationalized that since CSF-1R is highly expressed in tumour-infiltrating macrophages, such compounds may be useful in cancer imaging beyond Trk. We synthesized fluorinated derivatives and performed labelling of the most promising lead, 5-(4-((4-[^18^F]fluorobenzyl)oxy)-3-methoxybenzyl)pyrimidine-2,4-diamine ([^18^F]**3**, Figure 3). While inhibitor [^18^F]**3** maintained the excellent kinome selectivity of GW2580, its radiosynthesis was challenging and relied on the synthesis of the prosthetic group 4-[^18^F]fluorobenzyl bromide [55] followed by subsequent reaction with a phenolic labeling precursor. The multistep procedure only produced low radiochemical yields (RCYs) of [^18^F]**3**. This technical shortcoming precluded this tracer from its translation into a clinical setting and only most recent efforts applying Cu-mediated late stage fluorination techniques provided this tracer in high RCYs and high molar activities suitable for human PET scanning (unpublished). More recently, the quinazoline-based type II pan-Trk inhibitor [^18^F]QMICF ([^18^F]**4**) was developed (Figure 3) [53]. Starting from the corresponding racemic quinazoline-based dual FLT3/TrkB inhibitor, a structure–activity relationship study demonstrated that the (*R*)-enantiomer was primarily responsible for Trk inhibition [56]. To conserve optimal target interaction with the Trk receptor, the position of fluorine introduction was carefully consolidated through molecular modeling. Only one particular position, namely the allosteric pocket fragment, proved adaptable towards structural modifications with the least detrimental effect on binding interaction. We replaced the isopropyl moiety connected in 4-position to a structurally flexible benzene moiety in the original inhibitor with a 2-fluoro ethyl group to facilitate an easy introduction of ^18^F^−^ via common nucleophilic substitution of a corresponding mesyl derivative. This modification led to an about 10-fold potency reduction but also favorably abrogated FLT3 activity. Kinases profiling of inhibitor **4** revealed excellent selectivity and a Calcein-AM assay in P-gp overexpressing MDCKII cells confirmed only low interaction with P-gp. Radiotracer [^18^F]QMICF could be straightforwardly synthesized from the corresponding mesyl precursor in RCYs of 18% in one step. The compound is currently being evaluated for Trk brain imaging and tumor imaging in TrkA-TPM3 (IC_50_ = 162 nM) overexpressing colon carcinoma KM12 mice tumor xenografts and other neoplasms bearing diverse *NTKR* fusions.

#### 2.1.3. Imidazo[1,2-*b*]pyridazine-Based Radioligands

In 2014, when we started the syntheses of imidazo[1,2-*b*]pyridazine-based radioligands, it was demonstrated that *NTRK* fusion were actionable drivers in human cancers, however only a few non selective Trk inhibitors had been used in a therapeutic approach. The Trk inhibitor landscape was however rapidly moving from exploratory tool inhibitor (such as GW441756) to more relevant drug leads and there was a clear trend in preclinical work and patent literature towards what has now become one of the most explored chemotypes, specifically imidazo[1,2-*b*]pyridazine and pyrazolo[1,5-a]pyrimidine-based inhibitors (Figure 4). Cognizant of this trend, and expecting that scaffolds selected for clinical oncological studies should display overall excellent druglike properties which could in turn be advantageous for achieving brain and peripheral imaging using PET, we reoriented our effort towards inhibitors of these classes. We found that in comparison to all potential radiotracers described above, the type I pan-Trk inhibitors sharing the (2-pyrrolidin-1-yl)imidazo[1,2-*b*]pyridazine structural motif (Figure 5), share exceptionally high affinities to all three Trk subtype receptors orders of magnitude higher than any other inhibitor described so far.^57^ Based on extensive molecular docking studies, a comprehensive PET-oriented library of imidazo[1,2-*b*]pyridazine-based compounds (bearing positions amenable for labeling using carbon-11 and/or fluorine-18) was reported in 2015 [57]. All new pan-Trk inhibitors were subjected to a [γ-^33^P]ATP-based enzymatic assay to assess activities towards the different Trk subtypes. Remarkably, eight inhibitors from this library displayed <200 pM potency against TrkB/C and were initially regarded as potential candidates for radiotracer development. Two structures, (±)-IPMICF6 and (±)-IPMICF10 (Figure 5) lent themselves towards an easy introduction of ^18^F via simple nucleophilic substitution and were translated into the corresponding ^18^F-radiotracers. Both compounds displayed favorable in vitro activity and kinome selectivity. Their ^18^F labeling was straightforward and yielded the corresponding radiotracers in 3% and 8% RCY, respectively. In autoradiography experiments, both compounds were conclusively apt to topologically map the known brain distribution (cortex, striatum, thalamus and cerebellum) of TrkB/C receptors in coronal rat brain sections in accordance with mRNA and protein levels. In an important follow up study, one inhibitor from the reported imidazo[1,2-*b*]pyridazine-based library, IPMICF16, was chosen for radiotracer translation based on the alignment of favorable physicochemical properties (as described above) and optimal pharmacological parameters [58]. Pan-Trk inhibitor IPMICF16’s structure is geared towards simple ^11^C-methylation radiolabeling. In order to confirm that (*R*)-IPMICF16 was the more active and hence more suitable enantiomer for radiotracer development, as expected based on crystal structure and docking analyses, both the *R* and *S* isomers were synthesized and evaluated. Non-radioactive (*R*)-IPMICF16 displayed IC_50_s of 4.0, 0.2 and 0.1 nM for human TrkA, TrkB, and TrkC, respectively and inhibitory constants (*K*_i_) of 2.80, 0.05 and 0.021 nM for TrkA, TrkB, and TrkC, respectively (200,000–500,000-fold over *K_m_* ATP for all Trks). The *S*-enantiomer was significantly less potent which is in line with our docking studies. Interestingly, this study showed that (*R*)-IPMICF16 constitutes a rare example where an ATP-competitive Trk inhibitor displays intra-Trk sub-type selectivity. With regard to TrkA, (*R*)-IPMICF16 showed 56-fold and 133-fold higher selectivity for TrkB and TrkC, respectively. It was further asserted that the compound shows high selectivity towards TrkB/C over >99% of the testable human kinome (369 targets). The radiosynthesis of [^11^C]-(*R*)-IPMICF16 was routinely achieved using the corresponding desmethyl-precursor and radioactive [^11^C]methyl iodide (or alternatively [^11^C]methyl triflate) in >10% RCYs. Soon thereafter, the robust synthesis was set up according to good manufacturing practices and regulations at four different production sites where [^11^C]-(*R*)-IPMICF16 was evaluated in mice, rats, non-human primates, and finally in one healthy human subject (Figure 6). 

Despite partial elimination of [^11^C]-(*R*)-IPMICF16 from the rat brain due to active efflux transporters in rodents (P-gp and breast cancer resistance protein) as demonstrated in double knockout *Mdr1a/b-Bcrp* mice, SUV after 60-min scan time was still 0.4. The brain TrkB/C specific binding of [^11^C]-(*R*)-IPMICF16 in vivo was unambiguously demonstrated by pharmacological challenge with the clinical pan-Trk inhibitor entrectinib. Up to 88% of the radioactivity signal could be blocked after intraperitoneal injection of entrectinib, albeit high blocking doses (350 mg/kg) were required due to the documented low brain uptake of this inhibitor. Encouraged by these promising data, an evaluation of [^11^C]-(*R*)-IPMICF16 was performed in non-human primates (NHP). In comparison to rodents, the brain kinetics in NHP was significantly slower with a continuously increasing radioactivity signal in the brain devoid of an observable washout throughout the entire 60 min PET scan. Regional analysis confirmed a heterogenous radioactivity distribution following Trk rich grey matter with highest uptake in thalamus (SUV 0.8). Radioactivity uptake in white matter was comparatively low, as expected and in accordance with low level Trk expression. Although the efflux kinetic profiles of [^11^C]-(*R*)-IPMICF16 was notable in rodents, data from NHP PET imaging indicated interspecies differences and suggested that efflux may not be a liability in higher species. Hence, it was decided to move the tracer forward towards human in vivo translation. This decision was further motivated by the demonstration that in vitro [^11^C]-(*R*)-IPMICF16 autoradiography was sufficiently sensitive to reproduce the known and significant region-specific hippocampal TrkB/C density reduction found in AD brains (compared to age-matched controls). All radioactivity signals in brain tissues (controls and AD) were blockable using a structurally unrelated pan-Trk inhibitor proving specificity of the observed uptake as a result of TrkB/C receptor engagement. Taken together, these preclinical data justified moving towards a first-in-human tracer evaluation with PET.

Injection of [^11^C]-(*R*)-IPMICF16 into a 41-year-old healthy male subject led to brain uptake of the radiotracer which was rapid, with a peak SUV in the thalamus of 1.5 after 25 s post injection, very similar to what was observed in NHP. After reaching equilibrium, the SUV in one of the highest Trk tissues, the thalamus, remained 0.7 after 60 min with low observable washout. In accordance with non-human primate PET data, the radioactivity distribution in human matched the topography of the Trk disposition in grey matter with lowest SUVs in TrkB/C low white matter regions (Figure 7). The overall distribution aligned affirmatively with the known Trk distribution from ex vivo studies with highest uptake in the thalamus, followed by cerebellum and cortex. No adverse effects were observed after the injection of [^11^C]-(*R*)-IPMICF16.

Owing to the detailed structure-activity study conducted in the first phase of this project, we intended to modify the structure of [^11^C]-(*R*)-IPMICF16 to enable labeling with fluorine-18 based on the recently introduced Cu catalysed late stage fluorination of non-activated aromatic boronic esters [59,60,61]. Despite several shortcomings such as the inevitable side formation of protodeboronation side products, consequently contaminating the ^18^F-labeled target molecule and therefore significantly reducing the effective molar activity (*A*_m_) (the ^18^F and H substitution product have very similar physicochemical as well as pharmacodynamics properties), the Cu-catalyzed ^18^F fluorination of boronic esters and the introduction of mesityl aryl iodonium salts as labeling precursors can be easily regarded as being among the most important developments of modern ^18^F-radiochemistry [62,63]. [^11^C]-(*R*)-IPMICF16 originally bore a methoxy group directly adjacent to the fluorine atom at the aromatic amide fragment (Figure 6) which was identified as one contributor for the overall P-gp liability observed. That methoxy group was removed resulting in a pan-Trk inhibitor with further improved subtype selectivity but slightly reduced affinity for TrkB/C to reach binding equilibrium faster than [^11^C]-(*R*)-IPMICF16. The efflux of the corresponding fluorinated inhibitor was also noticeably reduced in cells. This radiotracer, called [^18^F]TRACK (the R isomer), was conveniently obtained in one step via Cu mediated radio-fluorination of a corresponding boronic ester precursor (the Bpin derivative) in satisfactory RCYs for further translation [64]. During the ^18^F-fluorination, the protodeboronation product was formed which reduced the effective A_m_ to a low value of 15 GBq/µmol, an order of magnitude below what was expected. After extensive screening, this problem could be solved by employing a pentafluorophenyl (PFP) HPLC column for tracer purification—which constitutes currently the standard method used to achieve separation [65]. The column material showed improved stationary phase affinity interaction with the fluorinated TRACK tracer over standard octadecylsilane reversed phase, leading to a base-line separation of [^18^F]TRACK from all impurities—including the protodeboronation side product. This improved the *A*_m_ to >100 GBq/µmol. In vivo evaluation in rhesus monkey confirmed uptake of [^18^F]TRACK in TrkB/C-rich regions with SUVs in the grey matter ranging from 0.9 for the cerebellum, 0.8 for the thalamus, and 0.6 for the cortex (Figure 7). White matter uptake was comparably low (0.2 SUV). The brain kinetics of [^18^F]TRACK were significantly different than what has been observed for [^11^C]-(*R*)-IPMICF16, displaying characteristics of a more reversibly binding radiotracer. Injections of [^18^F]TRACK of varying *A*_m_ (low, medium and high) into the same monkey illustrated the paramount importance of high *A*_m_ for Trk PET imaging. The PET signal was obviously reduced in cases of sub-optimal *A*_m_. [^18^F]TRACK is currently evaluated in human healthy controls and first in human PET data will be reported in due course. First obtained results with [^18^F]TRACK in a healthy human volunteer are in line with the data obtained with [^11^C]-(*R*)-IPMICF16 in terms of PET image quality and ease of production (unpublished).

## 3. Conclusions

The analysis, investigation and quantification of Trk receptors in neurodegenerative diseases and cancer is currently limited to ex vivo post-mortem analysis or invasive methods. While these kinases constitute important molecular targets which can be visualized using non-invasive PET imaging, Trk PET imaging is still in its infancy. Whether prototypical radioligands such as [^11^C]-(*R*)-IPMICF16 and [^18^F]TRACK, which display moderate brain uptake in humans, will be suitable for clinical neuroimaging and can provide reliable CNS TrkB/C measurements or adequately detect reductions in receptor density remain to be established. While tracers with higher brain penetration, volume of distribution (*V*_T_) values or binding potentials are desirable for temporally and spatially assessing Trk expression in conditions such as AD, the current tracers could certainly be beneficial in cases where Trk is overexpressed such as in numerous human cancers, both in the periphery and CNS. Further research is currently aimed at identifying structural determinants improving overall brain uptake of current type I and II tracers, including cyclic derivatives such as LOXO 195 and derivatives thereof. 

## Figures and Tables

**Figure 1 pharmaceuticals-12-00007-f001:**
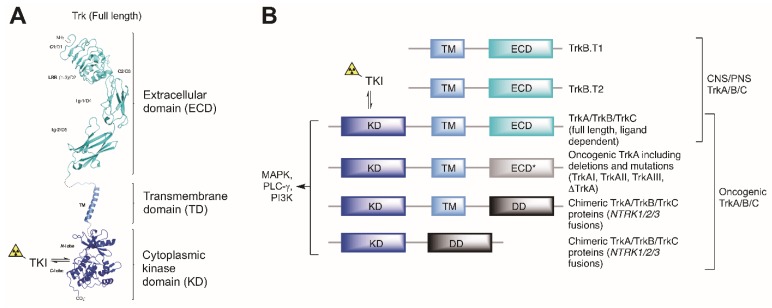
Detailed and representative domains of normally expressed and aberrantly expressed oncogenic tropomyosin receptor kinase (Trk) proteins from *NTRK* fusions (TKI: tyrosine kinase inhibitor). (**A**) Structure overview of the representative full TrkA receptor (D1–D5: domain 1–5; C1/3: cysteine cluster 1/3; LRR: leucine-rich repeat; Ig-1/2: immunoglobulin domain 1/2, TM: transmembrane domain). (**B**) Schematic representation of diverse Trk proteins and domains, including Trk splice variants and Trk fusion proteins. Dimerization and trans-autophosphorylation of Trk kinase domains leads to the activation of the downstream signaling pathways, including MAPK1, PI3-K/Akt and PLC-γ1 (DD: dimerization domain).

**Figure 2 pharmaceuticals-12-00007-f002:**
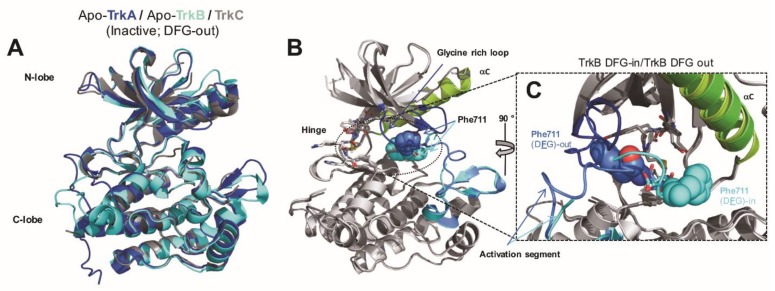
Trk kinase domain. (**A**) Overlap of TrkA, TrkB, and TrkC kinase domain (inactive conformations, PDB ID: 4F0I, 4ASZ, 3V5Q). (**B**,**C**) Views of the conformational differences between “Asp-Phe-Gly” DFG-in and DFG-out TrkB. The Phe residues of the DFG triad are shown in spheres (PDB ID: 4AT3, 4AT5).

**Figure 3 pharmaceuticals-12-00007-f003:**
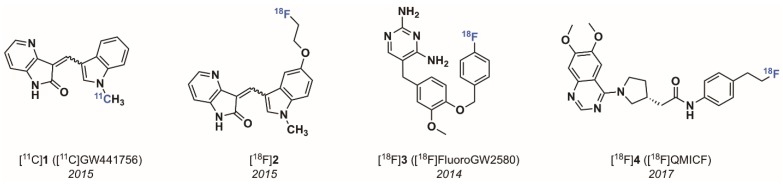
Chemical structures of early Trk-targeted positron emission tomography (PET) radioligands.

**Figure 4 pharmaceuticals-12-00007-f004:**
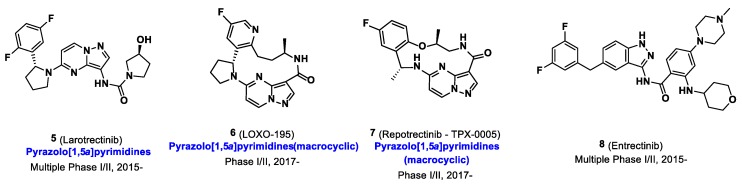
Chemical structures of selected clinical Trk inhibitors.

**Figure 5 pharmaceuticals-12-00007-f005:**
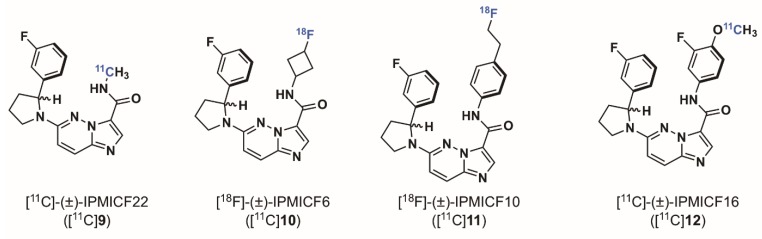
Chemical structures of preclinical imidazo[1,2*b*]pyridazine-based Trk-targeted PET radioligands.

**Figure 6 pharmaceuticals-12-00007-f006:**
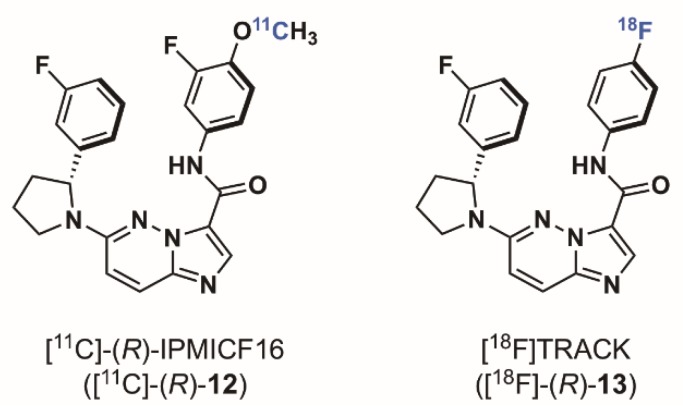
Chemical structures of clinical imidazo[1,2*b*]pyridazine-based Trk-targeted PET radioligands.

**Figure 7 pharmaceuticals-12-00007-f007:**
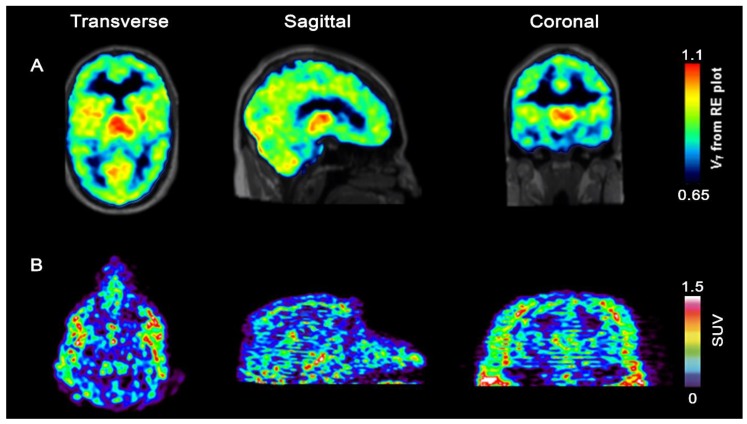
Upper row: PET/T1MP-RAGE MRT in vivo overlay images of [^11^C]-(R)-12 in a human subject with high SUVs in TrkB/C rich compartments such as thalamus, followed by cerebellum and cortical grey matter and low uptake in Trk devoid white matter areas; Bottom row: In vivo PET images of [^18^F]-(*R*)-13 (high *A_m_* of 245 GBq/µmol) in rhesus monkey brain matching the expected TrkB/C distribution.

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
