# Peer review of "Radioligands for Tropomyosin Receptor Kinase (Trk) Positron Emission Tomography Imaging"

_pharmaceuticals, 2019, doi:10.3390/ph12010007_

Round 1
Reviewer 1 Report
In the review article ‘Radioligands for Tropomyosin Receptor Kinase (Trk) 2 Positron Emission Tomography Imaging’ by R. Schirrmacher et al., the authors have produced an astonishingly thorough review of the history of radiotracer generation for TrkB for brain research (and for oncogenic research). This well referenced and thoughtful review is timely and will no doubt be a tremendous resource to the research community. The six figures provide excellent summary materials that will also be useful for didactic purposes.
I found the review to be a ‘good read’, as the authors provide a convincing narrative of the advances and the pitfalls of radiotracer development for Trk receptors for live imaging in humans and relevant animal models.
Minor comments are presented to improve the breadth and depth of the submission.
1. In the Introduction, the authors are encouraged to give a little more detail on the genetic structure of the Trk receptors, especially TrkB. There are several primary papers and/or reviews by Dr. Kaplan or by Dr. Moses V. Chao that provide this information.
2. The specific molecular mass of each TrkB isoform would be useful information to add to the manuscript. This could be added to Figure 2B.
3. Discourse on whether (or not) TrkB deficits in AD are early stage or late stage events would be helpful, and provides strong relevance for the development of tracers that can track the progression of dementing illness. There is a solid body of research indicating that BDNF and TrkB are downregulated early in the disease (e.g., in mild cognitive impairment- MCI) and correlate with cognitive decline, which would be appropriate to highlight in this comprehensive review.
Author Response
My colleagues and I are grateful for your useful comments regarding our manuscript. According to your recommendations, we have included a small paragraph about the genetic structure of the Trk receptor as well as information about the molecular mass of the most abundant TrkB isoform (new citations included; points 1 and 2 raised by reviewer 1):
Page 2: lines 55-61
Full length TrkB is a 140kD transmembrane spanning protein, with an extracellular ligand-binding domain containing two cysteine clusters, leucine-rich repeats, and two immunogloblulin-like domains4. The intracellular domain encodes a tyrosine kinase domain that when activated transphosphorylates monomers of the TrkB dimer. When transphosphorylated, the Trks engage and phosphorylate their major substrates Shc, PLC-gamma1, and SNT/FRS25. The TrkB locus also encodes four variants generated by alternative splicing, of which the most abundant is the 90kD truncated TrkB.t1 isoform that lacks the kinase domain6.
Furthermore we added more information regarding Trk in early stages of AD as suggested by reviewer 1 (point 3):
Page 2: lines 68-76
Direct evidence for the involvement of alterations in BDNF/TrkB signalling in AD is supported by studies indicating that TrkB levels are profoundly decreased in the hippocampus, frontal and temporal cortex of patients with Alzheimer’s18, although apparently not in the parietal cortex19. Furthermore, progressive loss of TrkA, B and C in basal forebrain cholinergic nuclei is well correlated with the clinical progression of AD20. Treatment with agonists of the BDNF/TrkB system of transgenic mouse models of AD increases dendritic spines in the hippocampus and cortex, inhibits neuronal apoptosis and neurodegeneration, and improves spatial memory performance21-25. These findings support the use of agents that activate BDNF/TrkB for treating AD26.
Reviewer 2 Report
I would like to show some PET images of candidates. The reader will interest the PET images of the compounds in non-human prompted and of course first-in-human study.
I suggest the authors include the images of [11C]-(R)-IPMICF16 (human brain) and [18F]TRACK (non-human primate).
Author Response
Dear Reviewer,
My colleagues and I are grateful for your useful comment regarding our manuscript.
As recommended by reviewer 2, we included a new figure (Figure 7) showing human as well a NHP PET images.
Figure 7. Upper row: PET/T1MP-RAGE MRT in vivo overlay images of [11C]-(R)-12 in a human subject with high SUVs in TrkB/C rich compartments such as thalamus, followed by cerebellum and cortical grey matter and low uptake in Trk devoid white matter areas; Bottom row: In vivo PET images of [18F]-(R)-13 (high Am of 245 GBq/µmol) in rhesus monkey brain matching the expected TrkB/C distribution.